# Effects of Different Pre-Harvest Bagging Times on Fruit Quality of Apple

**DOI:** 10.3390/foods13081243

**Published:** 2024-04-18

**Authors:** Zidun Wang, Yuchen Feng, Hui Wang, Xiaojie Liu, Zhengyang Zhao

**Affiliations:** 1College of Horticulture, Northwest Agricultural & Forestry University, Yangling District, Xianyang 712100, China; w17835697813@163.com (Z.W.); fengyuchen0622@163.com (Y.F.); wanghui106451@nwafu.edu.cn (H.W.); liuxiaojie@nxu.edu.cn (X.L.); 2Liaoning Institute of Pomology, Yingkou 115009, China

**Keywords:** Ruixue apple, bagging time, pre-harvest bagging, fruit quality, volatile compounds

## Abstract

Pre-harvest bagging can improve fruit color and protects against diseases. However, it was discovered that improper bagging times could lead to peel browning in production. Using the Ruixue apple variety as the research model, a study was conducted to compare the external and internal quality of fruits bagged at seven different timings between 50 and 115 days after full bloom (DAFB). Our findings indicate that delaying the bagging time can reduce the occurrence of peel browning in Ruixue apples. Compared to the control, the special bag reduced the browning index by 22.95%. However, the fruit point index of Ruixue fruits increased by 65.05% at 115 DAFB compared to 50 DAFB when bagging was delayed. The chlorophyll content of Ruixue fruits in special bags generally increased and then decreased, with the highest chlorophyll content of Ruixue fruits in special bags at 90 DAFB, which was 26.02 mg·kg^−1^. When the bagging process was delayed, the soluble solids, total phenols, and flavonoids content in the fruits increased, while the number of control volatiles decreased by 10. After two years of testing, results show that using special fruit bags at 90 DAFB bagging can significantly improve the fruit quality of Ruixue apple.

## 1. Introduction

Pre-harvest bagging is a well-established cultivation technique [1]. Apple bagging protects the fruit from pests, diseases, and birds, and reduces the amount of insecticides and fungicides on the fruit surface [2]. However, improper bagging can cause browning of the fruit. Bagging causes destruction of the pericarp cell structure, resulting in browning on the surface of the apple pericarp, with the decrease in flavonoids and increase in triterpenoids being the main causes of peel browning [3]. Despite the current labor shortage and rising labor costs, the use of bio-high-fat film spraying instead of fruit bags is becoming a popular trend. Nevertheless, in terms of pest and disease control, no-bagging technology is still imperfect [4]. Therefore, fruit bagging will continue to be a prevalent practice for the foreseeable future.

Fruit bagging can affect the biosynthesis of pericarp anthocyanins, chlorophylls, and carotenoids, which can improve the color of the pericarp [5]. Additionally, bagging can reduce the internal quality of the fruit, and the shading effect after bagging can inhibit the fruit’s photosynthesis, which may lead to a decrease in the content of sugar acids, phenolics, and aroma substances in the fruit [6,7]. The pigmentation of the pericarp is regulated by light. Bagging followed by bag removal increases fruit sensitivity to light, promotes anthocyanin synthesis, and enhances fruit coloration [8]. The main objective parameters used to evaluate fruit color are the L* value (brightness), a* value (red/green), and b* value (yellow/blue). Bagging was found to increase the L* value of “Golden Crown” apples compared to non-bagged fruits, suggesting that it can enhance the brightness of fruits [9]. The pericarp serves as a natural barrier that protects the fruit and enhances its ability to resist adversity. The appearance quality of fruit is determined by factors such as peel color, structure, and size. Additionally, the thickness of the peel cuticle plays a crucial role in the respiration of the fruit, which is essential for maintaining its intrinsic quality [10,11,12,13].

Several studies have concluded that bagging can significantly increase the fruit’s soluble solids content by increasing respiration and ethylene concentration while reducing titratable acid content [14,15]. Bagging has been found to have both positive and negative effects on the biosynthesis of secondary metabolites in fruit. For example, in loquat fruit, total phenol and total flavonoid contents were reduced after bagging treatment [6], while in pear pericarp, bagging reduced the phenolic content [16]. On the other hand, bagging has also been shown to significantly enhance the content of soluble solids and phenolic compounds in pear fruit [17]. Fruit hardness is a crucial factor in determining fruit quality and storage cycle. Research has shown that bagging can significantly improve fruit hardness in apples and pomegranates [15,18]. Moreover, studies on peaches have demonstrated that bagging can enhance the synthesis of volatile compounds within the fruit [19].

During the initial stages of promotion, Ruixue apples were frequently enclosed in a double-layer tricolor bag that provided a light shading effect, enhancing the fruit’s appearance, while this practice often resulted in peel browning. The closer the fruit was to maturity, the greater the likelihood of browning. Browning typically begins in the peel at the peduncle pits and then spreads to the fruit surface during storage, eventually taking on a radial appearance [20,21]. Previous studies have primarily focused on the mechanisms and substances involved in peel browning, but have not adequately addressed effective measures for mitigating browning. The purpose of this study was to determine the optimal timing and type of bag for bagging Ruixue apples during production to reduce browning while maintaining high fruit quality. To achieve this, we conducted experiments over two years, analyzing the effects of different bagging times on both the external and internal quality of the fruits. The results will provide a theoretical basis and technical support for the production of Ruixue fruits with higher quality.

## 2. Materials and Methods

### 2.1. Plant Materials and Treatments

The test material consisted of Ruixue apple trees (CNA20151469.1) that were 5 to 6 years old and grown on dwarfing autogenous rootstock M26. The trees were arranged in a 1.5 by 4 m spacing and had a slender spindle shape. Pest and disease control, as well as water and fertilizer, were routinely managed. The experiment was conducted at the Bashui Apple Experimental Station of Northwest A&F University in Baishui County, Weinan City, Shaanxi Province from April 2021 to November 2022. The station was located at 35°02′ N, 109°06′ E, with an altitude of 908 m, an average annual rainfall of 578 mm, and an average annual temperature of 11.4 °C.

In 2021, 42 Ruixue plants were selected based on good ventilation and light conditions. Seven bagging times were determined according to the bloom date (April 12): 50, 60, 70, 80, 90, 100, and 115 days after full bloom (refer to Table 1). In 2022, based on the results of a comprehensive evaluation in 2021, 4 bagging times were determined during the full bloom period of April 10. These times were 80, 85, 90, and 95 days after full bloom. Two types of fruit bags were selected: double-layer tricolor bags (control) and specific bags. The specifications of the bags are shown in Table 2. Six trees were selected for each treatment, resulting in 6 biological replications. Two different types of fruit bags were randomly placed on each part of the tree, resulting in a total of 180 fruits for each treatment. The fruits were harvested uniformly 190 days after full bloom (18 October 2021, 16 October 2022). Ten fruits were randomly selected for sampling in each treatment. After sampling, they were immediately stored in liquid nitrogen for freeze-drying and later determination of chlorophyll, total phenolic, flavonoid, peroxidase, and volatile indexes. Three biological replications were selected for each index and stored in a −80 °C refrigerator.

### 2.2. Determination of External Quality

Fruit surface color: the selection process involved choosing 10 fruits from the outer periphery of the tree, and a Minolta CR-400 colorimeter (Konnica Minolt, Tokyo, Japan) was used to measure the L* (representing the brightness value; the larger the L* value, the higher the brightness value), a* (representing the red/green value; the larger the a* value, the redder the color), and b* (representing the yellow/blue value; the larger the b* value, the yellower the color) of the fruit surface. In total, 100 fruits were selected for each treatment for browning index, glossy index, fruit point index, and coloring index. Browning classification (Figure A1): Class 0 no browning; Class 1 browning area S, 0 < S ≤ 1/3; Class 2 browning area S, 1/3 < S ≤ 2/3; Class 3: browning area S, 2/3 < S ≤ 1; browning index (%) = [Σ (browning level × number of fruits of that level)/(total number of fruits investigated (100) × highest level)] × 100. Smoothness classification (Figure A2): Class 0 with delicate skin; Class 1 with slightly rough fruit surface; Class 2 with rough, slightly dark fruit surface; Class 3 with rough fruit surface, like unbagged fruit. Brightness index: Brightness index (%) = [Σ (Brightness level × number of fruits of this level)/(Total number of fruits investigated (100) × highest level)] × 100. Fruit point classification (Figure A3): Class 0 fruit lenticels have a very small degree of lignification; Class 1 fruit lenticels have a small degree of lignification; Class 2 fruit lenticels have a large degree of lignification; Class 3 fruit lenticels show a large degree of lignification and browning. Fruit point index: Fruit point index (%) = [Σ (fruit point level × number of fruits of that level)/(total number of fruits investigated (100 fruits) × highest level)] × 100. Coloration classification (Figure A4): Class 0 fruit surface green or yellow uniform, no red; Class 1 fruit surface 1 cm^2^ with red; Class 2 fruit surface 3 cm^2^ with red; Class 3 fruit surface 1/4 area with red. Color index: color index (%) = [Σ(color grade × number of fruits of that grade)/(total number of fruits investigated (100) × highest grade) ] × 100. Chlorophyll determination: concerning the acetone colorimetric method of Zude–Sasse [22]. Each treatment involved selecting 10 apple peels, with three replications for the determination. Fruit shape index: longitudinal and transverse diameters of fruits were measured using vernier calipers, and fruit shape index was the value of longitudinal diameter/transverse diameter; 30 fruits were measured in each treatment. Individual fruit weight: the fruit was placed on an electronic balance for weighing, and 30 fruits were measured in each treatment.

### 2.3. Determination of Internal Quality

Fruit hardness was determined on three sides of the fruit using FTA-GS-15 Fruit Texture Analyzer (Mingao, Nanjing, China), measuring 20 fruits in each treatment. Soluble solids: determined by the ATAGO digital brix meter (Atago, Tokyo, Japan), 20 fruits were measured in each treatment. The titratable acid was determined by FRUIT ACIDZTY METER GMK-835F Apple Acidometer (GMK-835F Perfect, Berlin, Germany), and 20 fruits were measured in each treatment. Determination of antioxidant substances: 1.0 g of peel was weighed, ground in liquid nitrogen, and 1.5 mL of ethanol–acetone mixture (7:3, *v*/*v*) was added and incubated at 37 °C for 1 h. The homogenate was centrifuged at high speed for 20 min at 20 °C at 8000× *g*, and the supernatant was taken and stored immediately at −20 °C for the determination of total phenols, flavonoids, and total flavonoids. The total phenol, total flavonoid, and total flavonoid contents were determined by the method of Zhang et al. [14].

POD enzyme activity: the method of Wang et al. [1] was referred to for the determination of OD value by colorimetry at a wavelength of 470 nm. Volatile aroma substances of fruits: Referring to the method of Liu et al. [23], the fruits were extracted by headspace solid-phase microextraction using Trace DSQ GC/MS gas chromatography–mass spectrometry, and the ratio of the peak area of each component to the peak area of the internal standard was analyzed semi-quantitatively by using 3-nonanone as the internal standard, and the relative contents were expressed as the percentages of the substance contents of each component to the total substance contents.

### 2.4. Assessment of the Membership Function

Regarding the browning index, glossy index, fruit point index, coloring index, and other indicators, the smaller the value the better; for the inverse indicator, the independent indicator values for the dimensionless processing were bi (i = 1,2,3, …, n) for the corresponding indicators of the specific data. i is a different indicator, with the largest value of b (max) and the smallest value of b (min). Calculation was done using the following formula:(1)$$D\left(bi\right)=\frac{b(max)-bi}{b\left(max\right)-b(min)}$$

The positive indicator is calculated as:(2)$$D\left(bi\right)=\frac{bi-b\left(max\right)}{b\left(max\right)-b(min)}$$

Comprehensive evaluation of quality: The internal quality and external quality scores are each ranked at 50% for a composite score.

### 2.5. Statistical Analysis

Three biological replications were conducted to determine the experimental data. The data were organized using Microsoft Office Excel 2010 software and expressed as mean ± standard deviation. ANOVA and correlation analysis were performed using SPSS 26.0 software. Duncan’s statistical method was used to calculate the significance (*p* < 0.05) among treatments in ANOVA. The same type of fruit bag in the same year represents one treatment. Bivariate Pearson correlation analysis was used for correlation analysis, and squared Euclidean distance was used for clustering analysis to calculate the distance d value, which represents the similarity among treatments. The smaller the value of d between treatments, the closer the similarity.

## 3. Results

### 3.1. Effect of Different Bagging Times on the External Quality of Ruixue Apple

#### 3.1.1. Effect of Different Bagging Times on the Apparent Quality of the Peel of Ruixue Apple

Based on the photos in Figure 1, there are noticeable differences in the surface finish and fruit spot size of Ruixue fruit after treatment at different bagging times. With special bags, the fruit surface appeared greener when treated 90 days after full bloom, while the fruit treated at 100 and 115 days after full bloom displayed a yellowish-green color. The use of double-layer tricolor bags produced a yellow color, whereas special bags resulted in a green coloration.

#### 3.1.2. Effects of Different Bagging Times on the Peel Browning of Ruixue Apple

The investigation of the peel browning of Ruixue apple at maturity in 2021 revealed that both bags exhibited a decrease in browning index and browning rate with a delay in bagging time. The browning index of the pericarp of double-layer tricolor bags ranged from 18.18% to 78.30% (mean value 45.33%) (Figure 2A). The browning index of the special bags ranged from 10.00% to 34.56% (mean 22.38%), which was 22.95% lower than the mean value of the browning index of the double-layer tricolor bags. In 2022, the browning index of the double-layer tricolor bags ranged from 24.36% to 37.77% (mean 29.24%) (Figure 2B). The browning index of special bags ranged from 2.44% to 14.18% (mean value 8.37%). The treatment of double-layer tricolor bags declined initially from 80 to 90 days after full bloom, followed by an increase at 95 days. This trend is similar to the trend observed from 80 to 100 days after full bloom in 2021. The results of both years’ experiments show that the special bags could reduce Ruixue peel browning compared to double-layer tricolor bags. Furthermore, delayed bagging was effective in slowing down fruit browning, and the use of specialized bags 90 days after bloom resulted in the lowest browning.

#### 3.1.3. Effects of Different Bagging Times on the Peel Fruit Spot, Coloration, and Smooth Index of Ruixue Apple

According to Figure 3A,B, in 2021, both types of bags had the highest fruit point index at 115 days after full bloom and the lowest at 50 days after full bloom. In 2022, both types of bags had the lowest index at 80 days after full bloom and the highest at 90 days, which is consistent with the results of the 2021 experiment. As depicted in Figure 3C,D, the peel coloration index of the double-layer tricolor bag was 0 in both 2021 and 2022. As shown in Figure 3E,F, in 2021, the lowest peel smooth index was recorded in the double-layer tricolor bag treatment at 50 days after full bloom, and the highest was recorded in the dedicated bag at 115 days after full bloom bagging. The two-year experiment data show that late bagging would increase the fruit point on the fruit skin surface and reduce the degree of fruit surface finish.

#### 3.1.4. Effect of Different Bagging Times on the Size, Shape, and Surface Color of Ruixue Apple

Table 3 shows that in 2021 and 2022, the different lengths of treatments of double-layer tricolor bags and special bags, single fruit weight, transverse diameter, longitudinal diameter, fruit shape index, and other indicators of change did not yield significant differences. In 2021, double-layer tricolor bag treatment was used and color was measured using L* and a* values, with the delay of the time of bagging gradually reducing the values. The trend of L* value related to special bags was consistent with that of double-layer tricolor bags, which gradually decreased with the delay of bagging time. The a* value was significantly lower than that of other times at 70 to 100 days after full bloom, and the b* value increased with the delay in bagging time. According to the color measurement of double-layer tricolor bags in 2022, the L* value showed a decreasing trend with the delay of bagging time. The results of two years of testing show that early and late bagging had no significant effect on the fruit size and shape; for later bagging, the L* value (brightness value) is lower while the b* value (yellow/blue value) is higher, which indicates that the late bagging will reduce the brightness of the fruit surface, and increase the yellow degree of the fruit surface color.

#### 3.1.5. Effects of Different Bagging Times on the Peel Chlorophyll Content of Ruixue Apple

Determination of chlorophyll *a*, chlorophyll *b* and total chlorophyll content in Ruixue peel showed (Table 4) that among the treatments with double-layer tricolor bags in 2021, the peel chlorophyll *a* content of the bagging treatments at 100 and 115 days after full bloom were significantly higher than the others, whereas there was no significant difference between the chlorophyll *b* content and the total chlorophyll content. The three chlorophyll contents of the special bags were significantly different, with chlorophyll *a* content being the lowest for the bagging at 60 days after full bloom and the highest for the bagging 90 days after full bloom. The highest chlorophyll *b* content was found at 90 days after full bloom, which was 1.25 times higher than that of the treatment at 115 d days after full bloom. Comparison of the total chlorophyll contents of the special bags revealed that, at 80 and 90 days after full bloom, they were significantly higher than in several other treatments, with a general trend of increasing and then decreasing. Combined with Table 3 and Figure 1, it can be seen that the treatments using a special bag at 80 and 90 days after full bloom yielded a greener peel color compared to the other treatments. In the test results of 2022, both chlorophyll *a* and total chlorophyll contents of the special bags were the highest at 90 days after full bloom, and the lowest at 95 days after full bloom, showing a trend of increasing and then decreasing, which is highly consistent with the results of the test in 2021. This indicates that the chlorophyll content of the pericarp was higher and the color of the pericarp was greener at 90 days after full bloom in the special bag.

### 3.2. Effects of Different Bagging Times on the Internal Quality of Ruixue Apple

#### 3.2.1. Effects of Different Bagging Times on the Soluble Solid Content, Titratable Acidity Content, and Hardness of Ruixue Apple

As can be seen in Figure 4A, the soluble solids content of double-layer tricolor bags at different bagging times in 2021 ranged from 14.26 to 15.63% (mean 14.72%), with higher results for bagging 100 and 115 days after full bloom, which were significantly higher than the others, at 15.62% and 15.55%, respectively. The values for specific bags ranged 13.86% to 15.37% (mean 14.46%), which values were significantly higher at 115 days after full bloom than at several other times (except the treatment at 100 days after full bloom), at 15.37%. As can be seen in Figure 4C, there was no significant difference in titratable acid content between the two fruit bag treatments at different bagging times. The measurement of fruit hardness showed (Figure 4E) that, in 2021, the hardness of double-layer tricolor bags at the seven bagging times ranged from 6.46 to 7.34 kg·cm^−2^, among which the hardness of fruit treated with double-layer bags at 80 to 115 days after full bloom was significantly higher than that at 50 to 70 days after full bloom. According to Figure 4F, the trend for 2022 is consistent with that for 2021. A comparison of the two types of bags showed that the hardness values of the special bag treatments were higher than those of the double-layer tricolor bags at the same time, at an average of 2.50% higher. The results of both years’ experiments shows that delayed bagging could increase the soluble solids content and hardness of Ruixue fruits.

#### 3.2.2. Effects of Different Bagging Times on the Total Phenolic Content, Flavonoid Content, and Peel Peroxidase Activity of Ruixue Apple

Figure 5A shows that the total phenol content of the fruit from the two bags varied greatly after treatments at different bagging times in 2021. The total phenol content of the fruit from the double-layer tricolor bags ranged from 0.48 to 0.60 mg·g^−1^. The lowest content was 0.47 mg·g^−1^ at 50 days after full bloom, while the treatment had the highest content (significantly higher than the other treatments) at 115 days after full bloom. The total phenol content of the special bag for Ruixue fruits varied from 0.51 to 0.68 mg·g^−1^, with the lowest total phenol content being 0.51 mg·g^−1^ at 50 days after full bloom. The content significantly increased to 0.68 mg·g^−1^ at 115 days after full bloom compared to others. As shown in Figure 5B, the 2022 experiment yielded results consistent with the 2021 trend. The total phenol content of special bags was higher than that of double-layer tricolor bags, and the phenol content in the fruit increased with delayed bagging time. According to Figure 5C,D, the total flavonoid content in the fruit of both bags increased as the bagging time was delayed in 2021 and 2022, and this trend was consistent with the trend of total phenol content mentioned above. Pericarp peroxidase (POD) activity measurements showed (Figure 5E) that both fruit bags showed an increasing and then decreasing trend in 2021, with an increase from 50 to 90 days after full bloom and a decrease from 90 d to 115 days after full bloom. In the 2022 experiment (Figure 5F), there was no significant difference in the pericarp POD activities of the two types of bags after the four bagging time treatments. Comparing the results of the two-year experiment, we found that the overall trend of pericarp POD activity in the seven bagging times in 2021 was first increasing and then decreasing, while the contents of total phenols and total flavonoids increased with the delay in bagging time.

#### 3.2.3. Effect of Different Bagging Times on the Number of Volatile Aroma Components of Ruixue Apple

The determination of volatile aroma compounds in the two types of bags showed that a total of 46 compounds were detected (Table A2 and Table A3): 24 esters, nine aldehydes, five alcohols, two acetones, five olefins, and one acid. According to Figure 6, the total numbers of aroma substance species in double-layer tricolor bags after different bagging time treatments in 2021 decreased with the delay in bagging time, which were 36, 35, and 34 species at 50 to 70 days after full bloom, respectively, and 30, 26, and 26 species at 90 to 115 days after full bloom, respectively. The special bags had the lowest number of aroma substances at 115 days after full bloom, which was 33, and the other bagging times showed not much difference. A comparison of the two types of bags showed that the number of aroma substances in special bags was higher than that in double-layer tricolor bags at the same time. The total number of aroma substance types in the double-layer tricolor bags in 2022 was 33. The number of substance types varied from 36 to 39 in the special bags, with the highest being bagged at 80 days after full bloom and the lowest at 95 days after full bloom. This suggests that, compared with the specific fruit bag with higher light transmittance, the double-layer tricolor bag with no light transmittance reduced the number of aroma substance species of Ruixue fruits, and the different bagging time had a greater effect on the aroma substance species of the double-layer tricolor bag, while the effect on the specific bag was smaller.

#### 3.2.4. Correlation Analysis of Different Bagging Times and Volatile Aroma Components

Pearson’s correlation analysis of the seven bagging periods in 2021 with volatile aroma compound species and relative contents (Figure 7A and Table A1) showed that the bagging periods of double-layer tricolor bags correlated with several indicators: the number of volatile aroma compound species (−0.966), hexyl hexanoate (−0.867), ethyl butyrate (−0.826), butyl butyrate (−0.854), propyl butyrate (−0.881), hexyl propionate (−0.873), butyl benzoate (−0.854) and n-hexanol (−0.782) were significantly negatively correlated, whereas amyl butyrate (0.921), 2-hexenal (0.849), cis-β-farnesene (0.758) and (4E)-1-methyl-4-(6-methylhept-5-en-2-ylidene)-cyclohexene (0.878) were significantly positively correlated (correlation coefficients in brackets). The bagging time of Ruixue specialty bags was significantly and negatively correlated with the following aroma volatiles (Figure 7B): hexyl hexanoate (−0.882), propyl hexanoate (−0.871), ethyl butanoate (−0.857), propyl butyrate (−0.930), hexyl propionate (−0.875), propyl 2-methyl butanoate (−0.936), 13-heptadecyne-1-alcohol (−0.793) and α-farnesene (−0.798). There was significant positive correlation with hexyl tiglate (0.858) and amyl butyrate (0.838). This suggests that the relative levels of eight substances, including α-farnesene and some esters, could be reduced by delaying the bagging period relative to the special bag. The comparison of the two bags in 2021 revealed significant negative correlations with hexyl hexanoate, ethyl butyrate, propyl butyrate, and hexyl propionate, and significant positive correlations with amyl butyrate (0.921) across bagging periods. This indicates that both fruit bags reduced the relative levels of four compounds, including hexyl hexanoate.

### 3.3. Comprehensive Assessment of the Quality of Ruixue Apple

The quality indexes measured include the browning index, smooth index, fruit point index, coloration index, total chlorophyll content, soluble solids content, hardness, total phenols content, total flavonoids content, peroxidase activity, and volatile aroma components. Formulas A and B were used to calculate the D(bi) value. The appearance and intrinsic qualities were each assigned a score of 0.5 and then added. The experiments conducted in 2021 and 2022 revealed that the special bags yielded the highest scores at 90 days after full bloom (Table 5, Table A4, Table A5, Table A6 and Table A7). This suggests that the combination of appearance and intrinsic qualities was best achieved using the special bags at this stage.

## 4. Discussion

### 4.1. External Quality of Ruixue Apples as Affected by Different Bagging Times

Ruixue peel browning is a physiological disease that is more likely to occur during the ripening period and can spread to the entire fruit surface in the late stage of the disease, which seriously affects its appearance and quality as well as the benefits to the grower, and is also an important constraint to its promotion and development at present. Wang et al. [21] concluded that the key gene for its browning is *MdLAC7*; the light-responsive transcription factor *MdHY5* binds to the promoter of *MdWRKY31* under light conditions and represses the expression of this gene, suggesting that light can inhibit the production of browning. In this study, the data from both years of the experiment showed that, compared with the double-layer tricolor bag with no light transmission, the special fruit bag with 25% light transmission could reduce the degree of browning of Ruixue peel to a greater extent, with an average reduction of 50.62% in 2021 and 71.37% in 2022. The reason for this may be that the expression of its browning gene *MdLAC7d* was suppressed in the light-permeable environment of the fruit, thus inhibiting the occurrence of browning; secondly, the results of both years of experiments in this study showed that the peel browning index could be reduced with the postponement of the time of bagging, which was probably a result of the combination of both temperature and light. The microdomain environment changed by bagging can affect the cell structure, stomatal size, and wax layer on the peel surface [24], which in turn affects the appearance quality of the fruit surface [25]. In this study, different bagging times had a greater effect on the fruit surface quality, and the results of both years of experiments show that the earlier the bagging, the higher the fruit surface quality.

Different conclusions have been reached regarding the effects of time of bagging on the longitudinal and transverse diameter of fruit and single fruit quality; Islam et al. [26] concluded that bagging increased the weight and diameter of mango fruit. It was also noted that apple and pear fruits after bagging at different times had lower fruit weights per fruit than un-bagged fruits, and the effects of bagging time on fruit weight per fruit were not significant [27]. Reports indicate that the differences in longitudinal and transverse diameters of pear fruits and fruit weight per fruit were not significant after bagging with triple bags as compared to other bags [28]. This is consistent with the results of the present study: the results of both the 2021 and 2022 trials show that there were no significant effects of different bagging times on single fruit weight, longitudinal and transverse diameters, and fruit shape index of Ruixue fruits, which results are different from the vast majority of studies on apples, and the reason for this may be due to the varietal as well as bagging differences. Ruixue apple is a yellow-green variety; its fruit surface color is mainly affected by chlorophyll content. Several studies have pointed out that bagging affects fruit exposure to light, which can alter pericarp chlorophyll metabolism, and the longer the fruit is in the bag, the lower the pericarp chlorophyll content [5,29,30]. In contrast, in the 2021 trial of this study, the trends of chlorophyll *a*, chlorophyll *b*, and total chlorophyll contents of special bags were increased from 50 d to 90 days after full bloom, while decreasing from 90 to 115 d. It can also be found in conjunction with Figure 6 that the fruit surface was greener from 50 to 90 days after full bloom, while it gradually changed to yellowish green at 90 days after full bloom. The changes of chlorophyll increasing and then decreasing may be due to the later bagging changing the maturity of the fruit, resulting in the earlier fading of the green color on the fruit surface. The results of both the 2021 and 2022 experiments show that the delay of bagging time decreases the L* (brightness value) and increases the b* (yellow/blue value) on the fruit surface.

### 4.2. Internal Quality of Ruixue Apples as Affected by Different Bagging Times

Changes in light and temperature within the bag may affect intrinsic fruit quality by affecting fruit metabolism. Although there are some reports suggesting that bagging increases fruit soluble solids content [31], most studies have shown that bagging decreases intrinsic fruit quality [32]. For instance, the study found that when “Yellow Crown” pears ripened, the overall soluble solids content was significantly lower than that of un-bagged fruit. Bagging may affect the light environment of the fruit, leading to dark conditions that inhibit photosynthesis and reduce soluble solids and titratable acid contents [33]. Late bagging was found to increase the soluble solids content of the fruit in this study. Both types of fruit bags were used in this study.

Several studies have shown that bagging significantly reduces the hardness of a variety of fruits such as apples and guavas [34,35]. In the present study on fruit hardness, the results of both 2021 and 2022 show that the longer the fruit developed in the bag, the lower the hardness of the fruit. In the present study, the hardness of fruits bagged at 10 days after full bloom was 1.68 kg·cm^−2^ lower than that of CK, whereas for those bagged at 50 days after full bloom, it was only 0.9 kg·cm^−2^ lower than that of CK. This may be because early or late bagging affects the activities of polygalacturonase (PG), p-galactosidase (p-Gal), and cellulase (Cx), which affect the metabolism of cell wall substances, leading to changes in fruit hardness [7,15]. In the present study, the results of both 2021 and 2022 experiments show that the time of bagging had a significant effect on the total phenolic and flavonoid content of Ruixue fruits, both of which showed an increase with bagging. This result also showed an opposite trend to the occurrence of browning. Gao et al. (2023) showed that the occurrence of browning in Ruixue was closely related to the pericarp flavonoid content, which was mainly characterized by the trend of a higher flavonoid content correlating with a lower the degree of pericarp browning [36]. Total phenolic content, as well as polyphenol oxidase activity and *MdPPO1* gene expression, were lower in browned peels compared to normal peels [20]. This may be because early bagging reduces the antioxidant enzyme activity and antioxidant content of the pericarp [37], which reduces the fruit’s resistance to adversity and triggers an imbalance in the pericarp cellular antioxidant system, exacerbating the development of browning [38].

In pear studies, it was concluded that POD activity plays a key role in lignin biosynthesis, catalyzing the polymerization of lignin, that POD activity was significantly lower in the peel of bagged fruit than in that of unbagged fruit, and that the reduction in lignin content may be related to the formation of browning spots on pear peel [39]. The general trend of peroxidase activity in 2021 in this study was that it increased from 50 to 90 days after full bloom in bags and decreased from 90 to 115 days after full bloom. Interestingly, both the 2021 and 2022 data show that the browning rate and browning index of Ruixue apple after triple bagging treatment displayed an increasing trend at 90 days after full bloom, followed by a decrease, which is consistent with the changes in POD activity, and therefore, the browning of Ruixue apple reflects a similar lignin biosynthesis process as that observed in previous studies [40].

A mixture of different aroma substances constitutes the unique flavor of Ruixue fruits, with relatively high levels of hexyl 2-methyl butyrate, 2-hexenal, and α-farnesene being important for its unique flavor [41]. Bagging can reduce the overall concentration of volatile aroma compounds in the fruit, as reported in the Hanfu apple study [42], and in addition, Feng et al. [43] concluded that bagging could reduce the number of aroma substance species in Ruixue fruits by 16. In the present study, in 2021, the contents of double-layer tricolor bags decreased with the delay of bagging, and bagging at 115 days after full bloom removed 10 aroma components compared with bagging at 50 d, mainly in the reduction of esters, aldehydes, and alcohols. Therefore, it was hypothesized that the earlier bagging treatment created an adverse environment for Ruixue apples, which caused the fruit to synthesize more hormones related to resistance to the adverse environment, and the number of transcription factors responding to ethylene was higher under the regulation of hormones, so it was hypothesized that the earlier bagging promoted the accumulation of ethylene in the fruit, which led to an increase in aroma substances, and higher ethylene content was also associated with browning. Ethylene content also had a greater relationship with browning, which was also consistent with the browning trend in the results of this study [44,45,46]. In this study, the number of aroma components in the special bag did not show a significant trend in both years, and the number was higher than that of the double-layer tricolor bag, which may be due to the higher light transmittance of this fruit bag, which improves the synthesis of aroma substances [47]. In the present experiment, the effect of early or late bagging on aroma compounds is an interesting result. Although many studies have shown that pre-harvest bagging affects the synthesis of aroma volatiles by influencing the exposure of fruits to light, there is no report on the effect of early or late bagging on volatiles, and the effect of early or late bagging on volatiles in the present study has not been reported yet [42,48,49]. However, there is no report on the effect of early or late bagging on volatiles. In this study, it was found that the earlier the bagging, the higher the content of aroma substances was anyway, and the reason for this phenomenon involves a variety of complex factors such as light, temperature, and air, which will be further verified later.

## 5. Conclusions

Delayed bagging can improve the soluble solids, hardness, total phenols, flavonoids, and other intrinsic quality indicators of Ruixue apple, while reducing the degree of peel browning. However, it may also reduce the fruit surface brightness and finish, increase the fruit point index, and lower its appearance quality. The impact on the size and shape of the fruit is minimal. The two-year experiment’s results indicate that the browning index, peel chlorophyll content, soluble solid content, phenolic content, and volatile aroma components of the special fruit bags were superior to those of the three-color bags. The comprehensive assessment has revealed that the bags’ overall quality was highest 90 days after full bloom. Therefore, it is recommended that bagging approximately 90 days after full bloom during the production process can significantly improve the quality of Ruixue apple.

## Figures and Tables

**Figure 1 foods-13-01243-f001:**
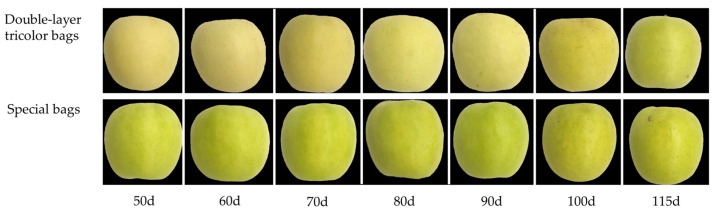
Effects of different bagging times on the external quality of Ruixue apple.

**Figure 2 foods-13-01243-f002:**
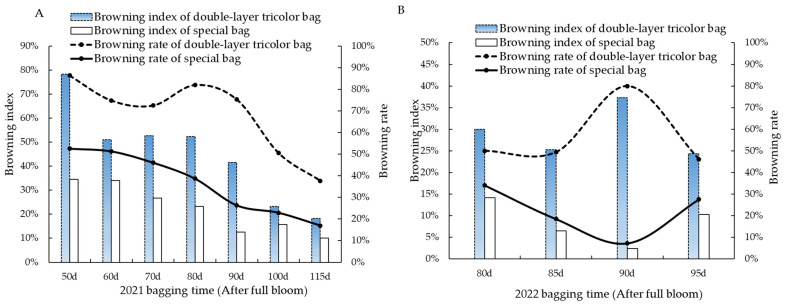
Effects of different bagging times on the peel browning of Ruixue apple. (**A**) Browning index and browning rate in 2021. (**B**) Browning index and browning rate in 2022.

**Figure 3 foods-13-01243-f003:**
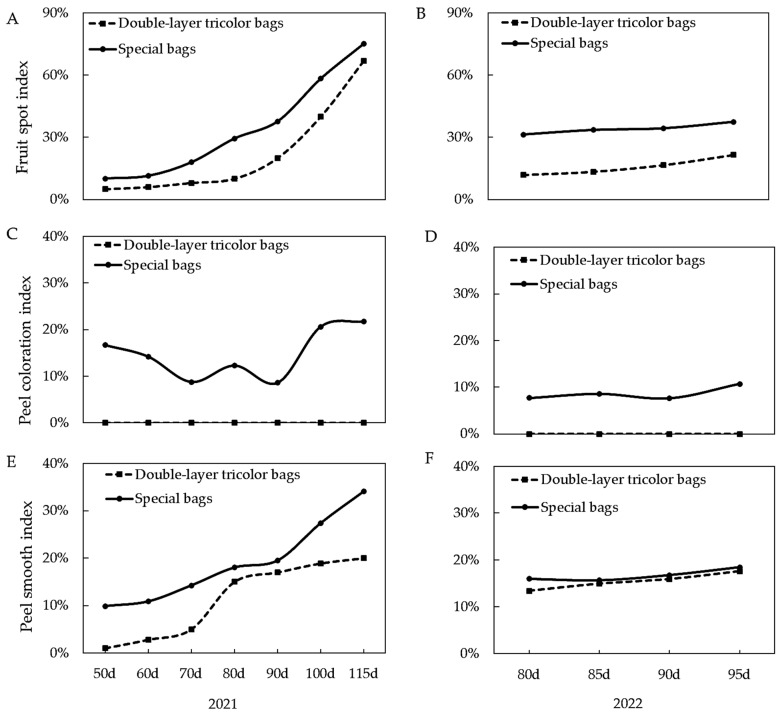
Effects of different bagging times on the peel fruit spot, coloration, and smoothness index of Ruixue apple. (**A**) Fruit spot index in 2021. (**B**) Fruit spot index in 2022. (**C**) Peel coloration index in 2021. (**D**) Peel coloration index in 2022. (**E**) Peel smooth index in 2021. (**F**) Peel smoothness index in 2022.

**Figure 4 foods-13-01243-f004:**
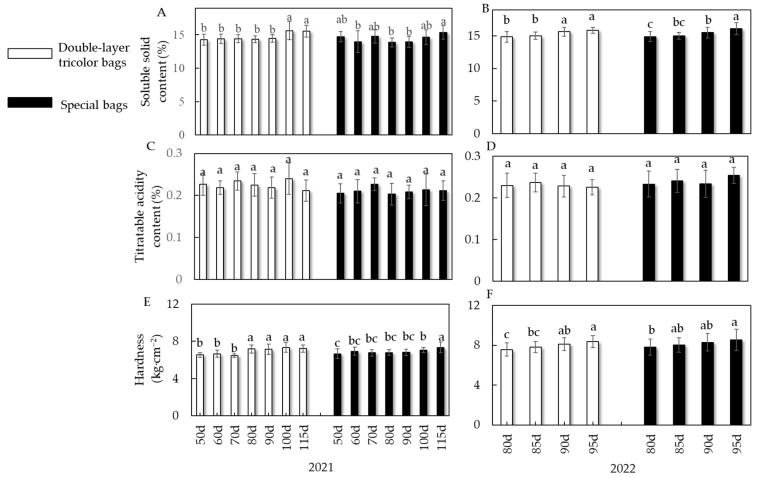
Effect of different bagging times on the soluble solid content, titratable acidity content, and hardness of Ruixue apple. (**A**) Soluble solid content in 2021. (**B**) Soluble solid content in 2022. (**C**) Titratable acidity content in 2021. (**D**) Titratable acidity content in 2022. (**E**) Hardness of fruit in 2021. (**F**) Hardness of fruit in 2022. Different letters indicate a significant difference (*p* < 0.05).

**Figure 5 foods-13-01243-f005:**
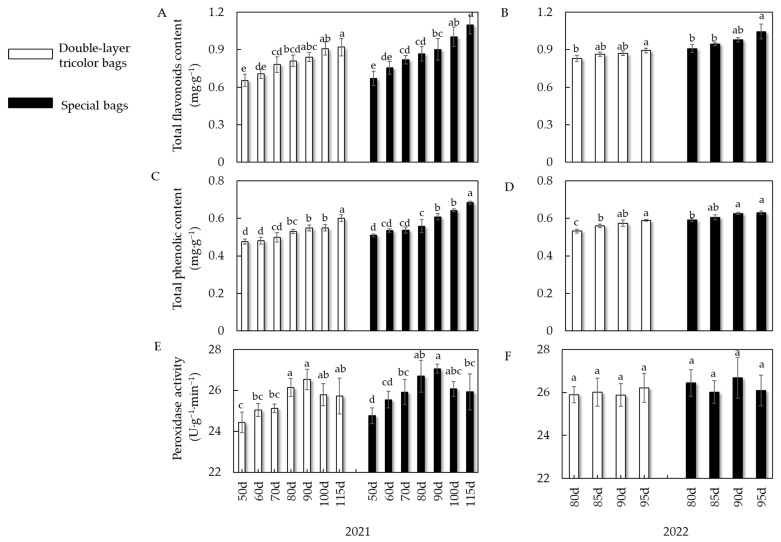
Effects of different bagging times on the total phenolic content, flavonoid content, and peel peroxidase activity of Ruixue apple. (**A**) Total flavonoids content in 2021. (**B**) Total flavonoids content in 2022. (**C**) Total phenolic content in 2021. (**D**) Total phenolic content in 2022. (**E**) Peroxidase activity in 2021. (**F**) Peroxidase activity in 2022. Different letters indicate a significant difference (*p* < 0.05).

**Figure 6 foods-13-01243-f006:**
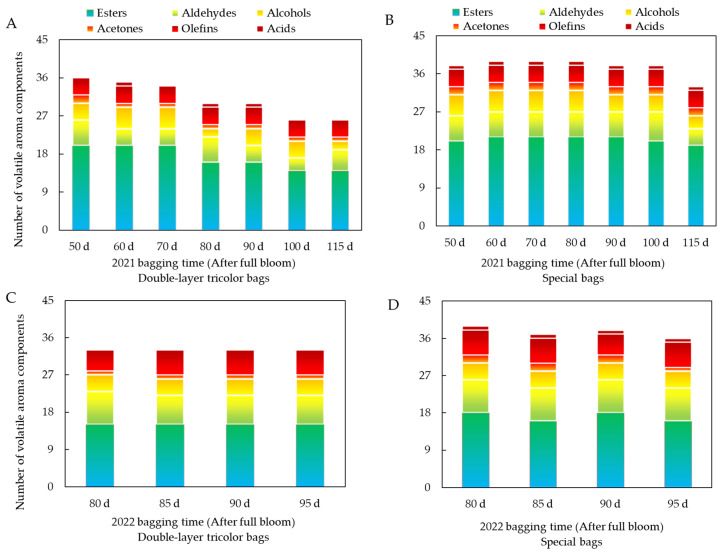
Effects of different bagging times on the number of volatile aroma components of Ruixue apple. (**A**) Number of volatile aroma components in 2021 for double-layer tricolor bags. (**B**) Number of volatile aroma components in 2021 for special bags. (**C**) Number of volatile aroma components in 2022 for double-layer tricolor bags. (**D**) Number of volatile aroma components in 2022 for special bags.

**Figure 7 foods-13-01243-f007:**
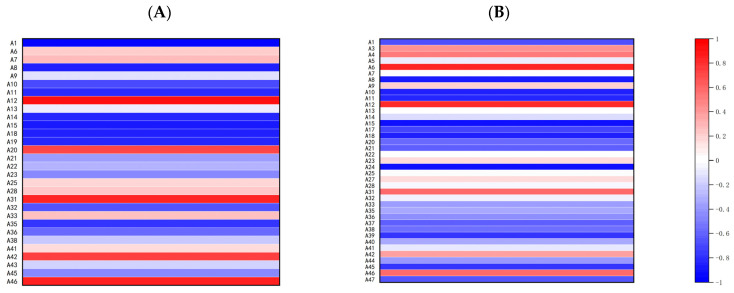
Correlation analysis of different bagging times and volatile aroma components. (**A**) Correlation analysis of different bagging times and volatile aroma components in 2021 for double-layer tricolor bags. (**B**) Correlation analysis of different bagging times and volatile aroma components in 2021 for special bags.

**Table 1 foods-13-01243-t001:** Fruit size and sunburn rate at different bagging times.

Years	Bagging Time	Longitudinal (mm)	Lateral (mm)	Sunburn Rate (%)
2021	50 days after full bloom (May 31)	28.65 ± 0.46	27.00 ± 0.59	0
	60 days after full bloom (June 10)	34.75 ± 0.70	34.51 ± 0.39	0
	70 days after full bloom (June 20)	42.01 ± 0.91	41.56 ± 1.04	0
	80 days after full bloom (June 30)	51.32 ± 1.33	50.87 ± 0.97	0
	90 days after full bloom (July 10)	53.15 ± 1.48	54.37 ± 1.25	0
	100 days after full bloom (July 20)	57.25 ± 1.92	59.86 ± 1.64	6
	115 days after full bloom (August 4)	59.36 ± 1.78	63.29 ± 2.03	21
2022	80 days after full bloom (June 28)	52.69 ± 1.37	50.18 ± 0.95	0
	85 days after full bloom (July 3)	54.06 ± 1.05	53.29 ± 1.67	0
	90 days after full bloom (July 8)	54.63 ± 1.64	55.27 ± 1.33	0
	95 days after full bloom (July 13)	55.48 ± 1.19	56.94 ± 1.26	5

**Table 2 foods-13-01243-t002:** Analysis of fruit bags material difference.

Fruit Bag Types	Manufacturer	Specification	Material	Light Transmittance (%)
Double-layer tricolor bags	Hongtai Fruit Bags Factory	150 × 179 mm 4.2 g	Outer paper: Paper with yellow outside and black inside Inner paper: Red waxy paper	0
Special fruit bags	Hongtai Fruit Bags Factory	155 × 180 mm 5.1 g	Outer paper: Whitewood pulp paper Inner paper: Green waxy paper	25

**Table 3 foods-13-01243-t003:** Effects of different bagging times on the size and shape of Ruixue apple. Effects of different bagging times on the external quality of Ruixue apple.

Fruit Bag Types	Years	Bagging Time	Fruit Weight (g)	Transverse (mm)	Longitudinal (mm)	Fruit Shape Index	L*	a*	b*
Double-layer tricolor bags	2021	50 d	258.86 ± 25.39 a	81.14 ± 2.81 a	76.13 ± 3.11 a	0.96 ± 0.04 a	81.29 ± 0.84 a	−5.25 ± 0.95 a	30.97 ± 1.45 d
		60 d	248.40 ± 25.63 a	81.11 ± 3.94 a	74.73 ± 3.63 a	0.91 ± 0.05 a	80.91 ± 1.09 a	−6.01 ± 0.85 ab	33.29 ± 1.86 c
		70 d	259.00 ± 20.18 a	81.90 ± 4.39 a	75.12 ± 3.24 a	0.91 ± 0.04 a	80.95 ± 1.02 a	−6.03 ± 0.65 ab	33.17 ± 2.04 c
		80 d	254.75 ± 26.02 a	79.37 ± 4.86 a	75.37 ± 3.41 a	0.94 ± 0.08 a	77.06 ± 0.96 b	−8.33 ± 0.71 c	39.60 ± 1.22 b
		90 d	249.61 ± 30.95 a	81.05 ± 2.60 a	73.63 ± 4.70 a	0.91 ± 0.04 a	76.38 ± 2.01 b	−8.21 ± 1.21 c	38.06 ± 2.05 b
		100 d	260.39 ± 23.11 a	82.31 ± 2.80 a	76.22 ± 2.30 a	0.94 ± 0.03 a	74.70 ± 1.97 c	−7.10 ± 2.00 bc	38.97 ± 1.59 b
		115 d	261.39 ± 29.41 a	81.18 ± 3.87 a	76.18 ± 3.92 a	0.95 ± 0.06 a	74.75 ± 1.40 c	−8.55 ± 2.19 c	41.55 ± 1.20 a
	2022	80 d	271.96 ± 41.52 a	82.69 ± 4.49 a	76.53 ± 4.17 a	0.93 ± 0.04 a	77.93 ± 0.87 a	−8.27 ± 0.63 a	36.87 ± 0.85 b
		85 d	261.84 ± 33.51 a	81.84 ± 3.88 a	77.18 ± 5.11 a	0.94 ± 0.05 a	77.59 ± 0.96 a	−8.13 ± 0.46 a	37.19 ± 1.13 ab
		90 d	262.54 ± 40.64 a	81.88 ± 5.16 a	76.89 ± 5.06 a	0.94 ± 0.05 a	76.74 ± 1.00 ab	−8.77 ± 0.98 a	38.57 ± 1.34 ab
		95 d	271.04 ± 39.27 a	82.73 ± 3.50 a	78.67 ± 4.37 a	0.95 ± 0.03 a	75.30 ± 0.90 b	−8.79 ± 0.99 a	39.06 ± 1.39 a
Special bags	2021	50 d	233.53 ± 35.35 a	79.98 ± 3.76 a	74.30 ± 5.13 a	0.93 ± 0.05 a	74.37 ± 1.35 a	−13.37 ± 1.29 a	42.49 ± 1.81 c
		60 d	247.17 ± 37.78 a	79.77 ± 3.31 a	76.02 ± 5.94 a	0.95 ± 0.07 a	74.42 ± 1.29 a	−14.04 ± 0.86 ab	42.84 ± 1.51 bc
		70 d	244.47 ± 26.23 a	80.50 ± 3.39 a	74.12 ± 3.82 a	0.92 ± 0.04 a	73.10 ± 1.71 b	−14.50 ± 0.87 b	41.48 ± 1.17 c
		80 d	255.97 ± 43.61 a	81.68 ± 4.80 a	76.84 ± 6.20 a	0.94 ± 0.05 a	72.67 ± 1.15 bc	−14.81 ± 0.87 b	42.13 ± 1.66 c
		90 d	255.70 ± 35.01 a	83.01 ± 5.33 a	76.23 ± 4.30 a	0.92 ± 0.05 a	71.95 ± 0.79 c	−14.47 ± 1.36 b	44.11 ± 1.74 ab
		100 d	270.93 ± 30.39 a	84.68 ± 4.30 a	79.89 ± 5.03 a	0.94 ± 0.06 a	72.02 ± 0.87 bc	−14.82 ± 0.58 b	44.04 ± 1.35 ab
		115 d	252.43 ± 32.17 a	82.82 ± 2.38 a	75.83 ± 5.36 a	0.92 ± 0.05 a	70.83 ± 0.81 d	−14.25 ± 1.07 ab	45.06 ± 1.78 a
	2022	80 d	297.24 ± 47.30 a	84.50 ± 4.20 a	81.14 ± 5.58 a	0.96 ± 0.04 a	72.94 ± 0.63 a	−14.88 ± 0.61 a	42.45 ± 0.78 b
		85 d	295.80 ± 35.77 a	84.57 ± 3.14 a	81.22 ± 5.33 a	0.96 ± 0.04 a	72.81 ± 0.37 a	−14.25 ± 0.87 a	42.78 ± 1.53 ab
		90 d	284.13 ± 31.89 a	83.47 ± 3.07 a	78.89 ± 5.16 a	0.94 ± 0.04 a	72.52 ± 0.83 ab	−14.50 ± 0.80 a	43.55 ± 1.07 ab
		95 d	294.91 ± 30.04 a	83.74 ± 2.68 a	81.34 ± 2.78 a	0.97 ± 0.04 a	71.59 ± 0.58 b	−14.69 ± 0.59 a	44.35 ± 0.91 a

Note: The interval for ANOVA was the same for each type of fruit bag in the same year. Different letters indicate a significant difference (*p* < 0.05).

**Table 4 foods-13-01243-t004:** Effects of different bagging times on the peel chlorophyll content of Ruixue apple.

Fruit Bag Types	Years	Bagging Time	Chlorophyll *a* Content (mg·kg^−1^)	Chlorophyll *b* Content (mg·kg^−1^)	Total Chlorophyll Content (mg·kg^−1^)
Double-layer tricolor bags	2021	50 d	7.83 ± 0.68 b	12.52 ± 1.43 a	20.35 ± 1.98 a
		60 d	7.89 ± 0.29 b	11.62 ± 1.08 a	19.50 ± 1.28 a
		70 d	8.07 ± 0.35 b	11.33 ± 1.00 a	19.40 ± 1.25 a
		80 d	8.68 ± 0.68 b	11.30 ± 0.77 a	19.98 ± 1.45 a
		90 d	7.80 ± 0.25 b	11.89 ± 0.76 a	19.68 ± 1.00 a
		100 d	10.12 ± 0.52 a	12.18 ± 1.55 a	22.30 ± 1.94 a
		115 d	9.71 ± 0.40 a	11.25 ± 0.59 a	20.95 ± 0.97 a
	2022	80 d	6.33 ± 0.73 b	8.22 ± 1.05 b	14.56 ± 1.78 b
		85 d	6.27 ± 0.28 b	7.46 ± 0.22 b	13.73 ± 0.49 b
		90 d	6.83 ± 0.53 b	9.99 ± 0.63 ab	16.81 ± 1.16 b
		95 d	8.03 ± 0.77 a	13.17 ± 3.23 a	21.2 ± 3.80 a
Special bags	2021	50 d	12.70 ± 1.91 c	14.84 ± 1.84 c	27.53 ± 3.73 c
		60 d	12.35 ± 1.25 c	16.22 ± 1.26 c	28.57 ± 0.18 c
		70 d	12.44 ± 1.07 c	17.35 ± 0.58 c	29.79 ± 1.40 c
		80 d	18.48 ± 1.19 a	24.69 ± 2.52 b	43.17 ± 1.38 b
		90 d	19.38 ± 1.37 a	29.72 ± 5.39 a	49.10 ± 6.69 a
		100 d	16.76 ± 1.54 ab	15.52 ± 1.61 c	32.28 ± 3.12 c
		115 d	14.33 ± 0.27 bc	13.18 ± 0.36 c	27.51 ± 0.58 c
	2022	80 d	9.93 ± 0.88 b	11.74 ± 1.46 a	21.66 ± 1.77 ab
		85 d	12.80 ± 0.29 a	12.77 ± 0.79 a	25.56 ± 1.00 a
		90 d	12.95 ± 1.07 a	13.07 ± 2.51 a	26.02 ± 3.5 a
		95 d	9.57 ± 0.13 b	10.00 ± 1.73 a	19.58 ± 1.84 b

Note: The interval for ANOVA was the same for each type of fruit bag in the same year. Different letters indicate a significant difference (*p* < 0.05).

**Table 5 foods-13-01243-t005:** Comprehensive assessment of the quality of Ruixue apple.

Years	Fruit Bag Types	Bagging Time	External Quality	Internal Quality	Comprehensive Score	Rank
2021	Double-layer tricolor bags	50 d	0.303	0.091	0.394	14
		60 d	0.333	0.134	0.467	11
		70 d	0.321	0.131	0.453	12
		80 d	0.291	0.221	0.511	9
		90 d	0.285	0.249	0.534	8
		100 d	0.287	0.286	0.573	3
		115 d	0.248	0.292	0.540	6
	Special bags	50 d	0.280	0.161	0.441	13
		60 d	0.292	0.211	0.503	10
		70 d	0.312	0.256	0.568	4
		80 d	0.318	0.259	0.578	2
		90 d	0.354	0.296	0.651	1
		100 d	0.184	0.355	0.540	7
		115 d	0.127	0.413	0.540	5
2022	Double-layer tricolor bags	80 d	0.328	0.001	0.329	8
		85 d	0.298	0.081	0.379	7
		90 d	0.256	0.150	0.406	6
		95 d	0.276	0.241	0.517	3
	Special bags	80 d	0.232	0.245	0.477	5
		85 d	0.275	0.223	0.498	4
		90 d	0.274	0.396	0.670	1
		95 d	0.125	0.397	0.522	2

Note: The D(bi) values of the table are used to assess the membership function.

## Data Availability

The original contributions presented in the study are included in the article, further inquiries can be directed to the corresponding author.

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
