# Peer review of "Effects of Different Pre-Harvest Bagging Times on Fruit Quality of Apple"

_foods, 2024, doi:10.3390/foods13081243_

Round 1
Reviewer 1 Report
Comments and Suggestions for Authors
Manuscript Foods-2945713 reports on the pre-harvest bagging of Ruixue apples and its effect on fruit quality. Three general comments on the content of the present study are: 1) since considerable work has been done on the principle of pre-harvest bagging of apples including the Riuxue variety has been already done (see Reference list), the authors should clearly state the novelty of their work. This statement should be included in the last part of the Introduction section, 2) in case where the work involves the preservation/maintenance of food quality, it is advisable that the authors run a sensory evaluation of the product including evaluation of taste, odor, texture, etc. given that sensory attributes are the ultimate criteria of acceptance of the product by consumers, 3) for comparison purposes the experimental design should have included the un-bagged apples as a control. With the control missing one cannot truly comprehend the value of the bagging principle. All three of the above are missing from the study. English also requires refinement.
Detailed comments:
l.27: add a reference (s) to this statement
l.38-39: if bagging causes all these problems to fruit quality, then why is it used ?
Fig. 1: is the preferred color of Ruixue apples yellow-green or green. Based on my experience, apples of the Golden yellow variety are yellow-green at the stage of optimum maturity
Fig 2: why is there a different pattern for the Browning index between years 2021 and 2022 ?
In the conclusion section a comparison should be made on the specific effect of the special bag vs. the tricolor bag
Based on the above, I recommend major revision and reconsideration of the manuscript depending on how the authors address my comments, especially my general comments
Comments on the Quality of English Language
Reviewer 2 Report
Comments and Suggestions for Authors
This paper aimed to determine the optimal bagging time for ‘Ruixue‘ apple fruits. The authors analyse the effects of different bagging times on the external and internal quality of the fruits. The article is written logically and clearly, with a lot of experimental data.
Authors should make changes and additions to the text.
- Lines 17, 18, and 19: It should be specified what type of Ruixue was used, for example, Ruixue fruits or Ruixue apple fruits.
- Line 51: Instead of [8, 9, 10, 11], it should read [8-11].
- Line 136: Was fruit hardness measured on one or both sides of the fruit?
- Lines 241 and 243, Table 4: The correct formatting for Chlorophyll a and Chlorophyll b is to italicize the letters 'a' and 'b', for example, Chlorophyll a and Chlorophyll b, instead of Chlorophyll a and Chlorophyll b.
Reviewer 3 Report
Comments and Suggestions for Authors
Generally, this study presents practical knowledge on improving the post-harvest quality of Apples by pre-harvest bagging. While it seems like a simple study, this study has high-value for industrial applications
Is n=180 for this study? If yes, why are there no statistical representations of significant differences in figures?
In the abstract, there were no quantitative discussions of the results of this study.
Round 2
Reviewer 1 Report
Comments and Suggestions for Authors
Authors have reasonably addressed my comments. Therefore, the revised manuscript may be accepted for publication.